# Codon-level information improves predictions of inter-residue contacts in proteins by correlated mutation analysis

**Etai Jacob[1,2], Ron Unger[1]\*, Amnon Horovitz[2]\***

[1]The Mina and Everard Goodman Faculty of Life Sciences, Bar-Ilan University, Ramat-Gan, Israel; [2]Department of Structural Biology, Weizmann Institute of Science, Rehovot, Israel

**Abstract** Methods for analysing correlated mutations in proteins are becoming an increasingly powerful tool for predicting contacts within and between proteins. Nevertheless, limitations remain due to the requirement for large multiple sequence alignments (MSA) and the fact that, in general, only the relatively small number of top-ranking predictions are reliable. To date, methods for analysing correlated mutations have relied exclusively on amino acid MSAs as inputs. Here, we describe a new approach for analysing correlated mutations that is based on combined analysis of amino acid and codon MSAs. We show that a direct contact is more likely to be present when the correlation between the positions is strong at the amino acid level but weak at the codon level. The performance of different methods for analysing correlated mutations in predicting contacts is shown to be enhanced significantly when amino acid and codon data are combined.

## Introduction

The effects of mutations that disrupt protein structure and/or function at one site are often suppressed by mutations that occur at other sites either in the same protein or in other proteins. Such compensatory mutations can occur at positions that are distant from each other in space, thus, reflecting long-range interactions in proteins (*Horovitz et al., 1994*; *Lee et al., 2008*). It has often been assumed, however, that most compensatory mutations occur at positions that are close in space, thus motivating the development of computational methods for identifying co-evolving positions as distance constraints in protein structure prediction (*Göbel et al., 1994*). These methods, which rely on multiple sequence alignments (MSA) of homologous proteins as inputs, will become increasingly more useful in the coming years owing to the explosive growth in sequence data. The output of methods for correlated mutation analysis (CMA) is a rank order of the pairs of columns in the alignment according to the statistical and/or physical signficance attached to the correlation observed for each pair. The various methods for CMA that have been developed in the past 15 years differ in the measures they employ for attaching significance to the correlations (*Livesay et al., 2012*; *de Juan et al., 2013*; *Mao et al., 2015*). Early measures include, for example, mutual information (MI) from information theory (*Gloor et al., 2005*) and observed-minus-expected-squared (OMES) in the chi-square test (*Kass and Horovitz, 2002*).

Statistically significant correlations in MSAs that do not reflect interactions between residues in contact, that is, false positives, can stem from (i) various indirect physical interactions and (ii) common ancestry. The extent of false positives due to the latter source is manifested in the large number of correlations between positions in non-interacting proteins that can be observed when the sequences of non-interacting proteins from the same organism are concatanated and subjected to CMA (*Noivirt et al., 2005*). Several methods for removing false positives owing to common ancestry were developed (*Pollock et al., 1999*; *Wollenberg and Atchley, 2000*; *Noivirt et al., 2005*; *Dunn et al., 2008*) but their

*For correspondence:
ron@biomodel.os.biu.ac.il (RU);
Amnon.Horovitz@weizmann.
ac.il (AH)

**eLife digest** Genes contain instructions to make proteins from building blocks called amino acids. The instructions are encoded in units called codons that each specify a single amino acid in the chain. A small mutation in a particular codon can change the amino acid found at the corresponding position in the protein. Some amino acids interact with other amino acids in the chain, thereby enabling the protein to adopt the three-dimensional shape it needs to work properly. Therefore, a mutation that affects one of these amino acids may have a large impact on the ability of the protein to work.

A mutation at one position in the protein may, however, have little effect if it is accompanied by a 'compensatory' mutation at another position. Such compensatory mutations are more likely to occur when the two positions in the protein are close to each other. To identify such mutations, the amino acid sequences of similar proteins from different organisms are aligned and compared.

A computational method called 'correlated mutation analysis' searches for pairs of positions in the alignment that display co-variation, i.e. where particular mutations at one position tend to be accompanied by certain mutations at the second position. These pairs are then ranked according to the strength of their correlation and those with the highest ranking are predicted to be in close contact. Such predictions are, however, far from perfect and can give false results.

Jacob et al. developed and tested a new technique of correlated mutation analysis by examining codon sequences as well as amino acid sequences. The rationale behind the technique relies on the fact that several different codons can encode the same amino acid, so that a mutation in a codon does not always change the amino acid it encodes. Therefore, a strong correlation at the amino acid level can be accompanied by a weak correlation at the codon level. In such cases the positions are more likely to be in contact than in cases where there is a strong correlation also at the codon level since the correlation can then be due to constraints at the DNA or RNA level.

Jacob et al. tested their approach using different methods for analyzing correlated mutations that were proposed in previous studies. This showed that the predictions obtained using both amino acid and codon data are significantly more accurate than those obtained by comparing amino acid sequences only. Future work will test whether combining amino acid and codon data can also be used to predict interactions between different proteins.

success in contact prediction using CMA remained limited. False positives due to the former source, that is, indirect physical interactions, can occur when, for example, correlations corrersponding to positions i and j that are in contact and positions j and k that are in contact lead to a correlation for positions i and k that are not in contact. Methods that remove such transitive correlations have been developed in recent years and include, for example, Direct Coupling Analysis (DCA or DI for Direct Information) (*Weigt et al., 2009*; *Morcos et al., 2011*), Protein Sparse Inverse COVariance (PSICOV) (*Jones et al., 2012*) and Gremlin's pseudolikelihood method (*Kamisetty et al., 2013*). These methods have been found to be very successful in identifying contacting residues (*Marks et al., 2012*) and they outperform earlier methods (*Mao et al., 2015*). Nevertheless, their accuracy, which is ~80% for the correlations in the top 0.1% (ranked by their scores), drops to ~50% for the top 1% (*Mao et al., 2015*). Given that the number of contacts in a protein with N residues is ~N (*Faure et al., 2008*), it follows that for proteins with, for example, 100 residues (i.e. with 4560 potential contacts between residues separated by at least 5 residues in the sequence) only about 25% of the contacts (i.e. 23 of the top 1% 46 predictions) will be identified by these CMA methods. In addition, these methods require large MSAs comprising thousands of sequences in order to perform well and such sequence data are not always available. Consequently, it is clear that much can be gained from further improvements in methods of CMA. Here, we describe the development and application of a new method for CMA that uses both amino acid and codon MSAs as inputs instead of relying exclusively on amino acid MSAs as done before. We show that contact prediction is improved in a meaningful manner when amino acid and codon information are combined.

## Results and discussion

The key premise underlying the method introduced here is that a correlation at the amino acid level between two positions is more likely to reflect a direct interaction if the correlation at the codon level for these positions is weak (*Figure 1*). In other words, it is assumed that cases of strong correlations at

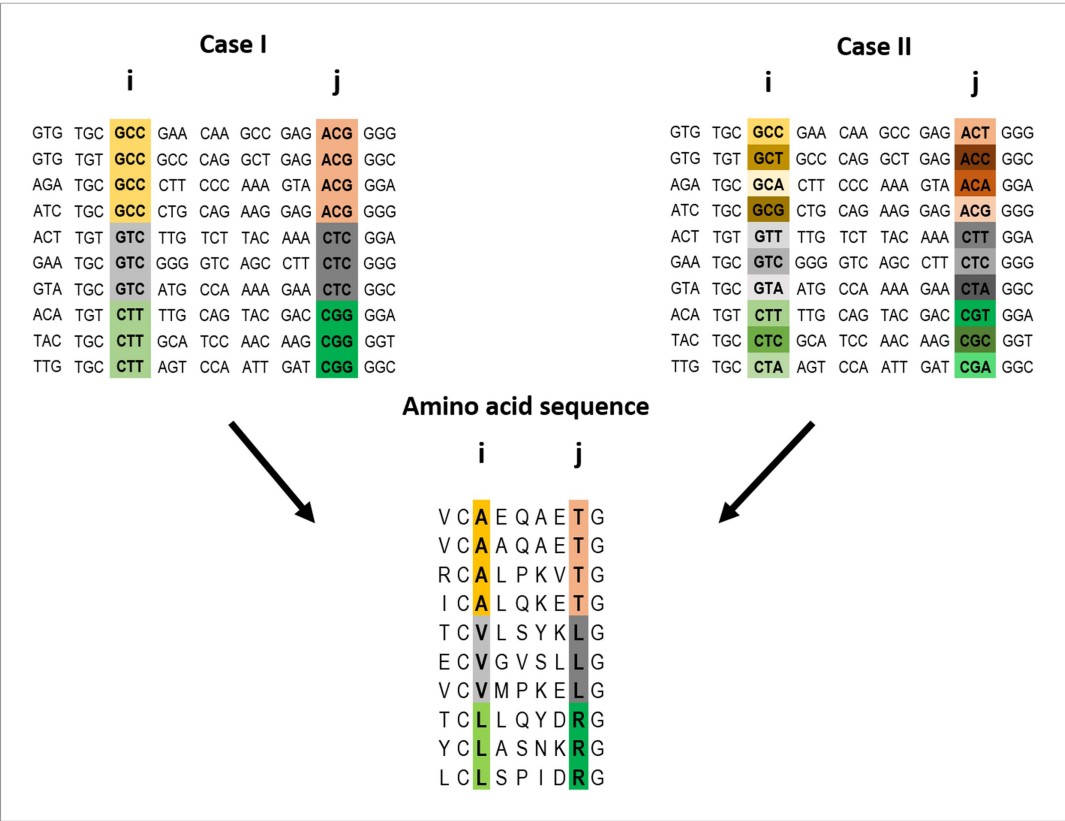

**Figure 1**. Example of a pairwise correlation in a multiple amino acid sequence alignment and two possible corresponding codon alignments. A correlation at the amino acid level between two positions i and j may (top left) or may not (top right) be accompanied by a correlation at the codon level. The premise of the method introduced here is that a correlation at the amino acid level between two positions is more likely to reflect a direct interaction if the correlation at the codon level for these positions is weak (top right).

both the amino acid and codon levels for a pair of positions are less likely to reflect selection to conserve protein contacts and more likely to reflect selection to conserve interactions involving DNA or RNA and/or common ancestry. Given this rationale in mind, we decided to test whether contact identification improves when all the pairs of positions are ranked using a score that increases with (i) increasing strength of the correlation at the amino acid level and (ii) decreasing strength of the correlation at the codon level. Such a score, $S_i$, is given, for example, by: $S_i = S_i^\alpha(AA)/S_i(C)$, where $S_i(AA)$ and $S_i(C)$ are the scores generated by method i (e.g., MI) for the amino acid and codon alignments, respectively, and the value of the power $\alpha$ is determined empirically depending on the method (see below).

The approach outlined above was tested for the OMES (*Kass and Horovitz, 2002*), MI (*Gloor et al., 2005*), MIp (*Dunn et al., 2008*) and DCA (*Morcos et al., 2011*) methods using 114 MSAs each comprising at least 2000 sequences of length between 200 and 500 residues. In the case of the PSICOV method (*Jones et al., 2012*), only 86 MSAs out of the 114 MSAs were used since the others didn't pass this method's threshold for amino acid sequence diversity. Each MSA also included at least one sequence with a known crystal structure at a resolution <3 Å in which at least 80% of all the residues are resolved. The mean accuracy of contact identification was plotted as a function of the top ranked number of predicted pairwise contacts (*Figure 2—figure supplement 1*) or as a function of the top ranked fraction of protein length, L, number of predicted pairwise contacts (*Figure 2*). Residues were considered as being in contact if the distance between their $C_\beta$ atoms is ≤8 Å following the definition used in CASP experiments (*Ezkurdia et al., 2009*) and other studies (*Kamisetty et al., 2013*; *Skwark et al., 2014*) (see also *Figure 2—figure supplement 2*). The results show that the PSICOV and DCA methods outperform the OMES, MI and MIp methods (*Figure 2*) as established

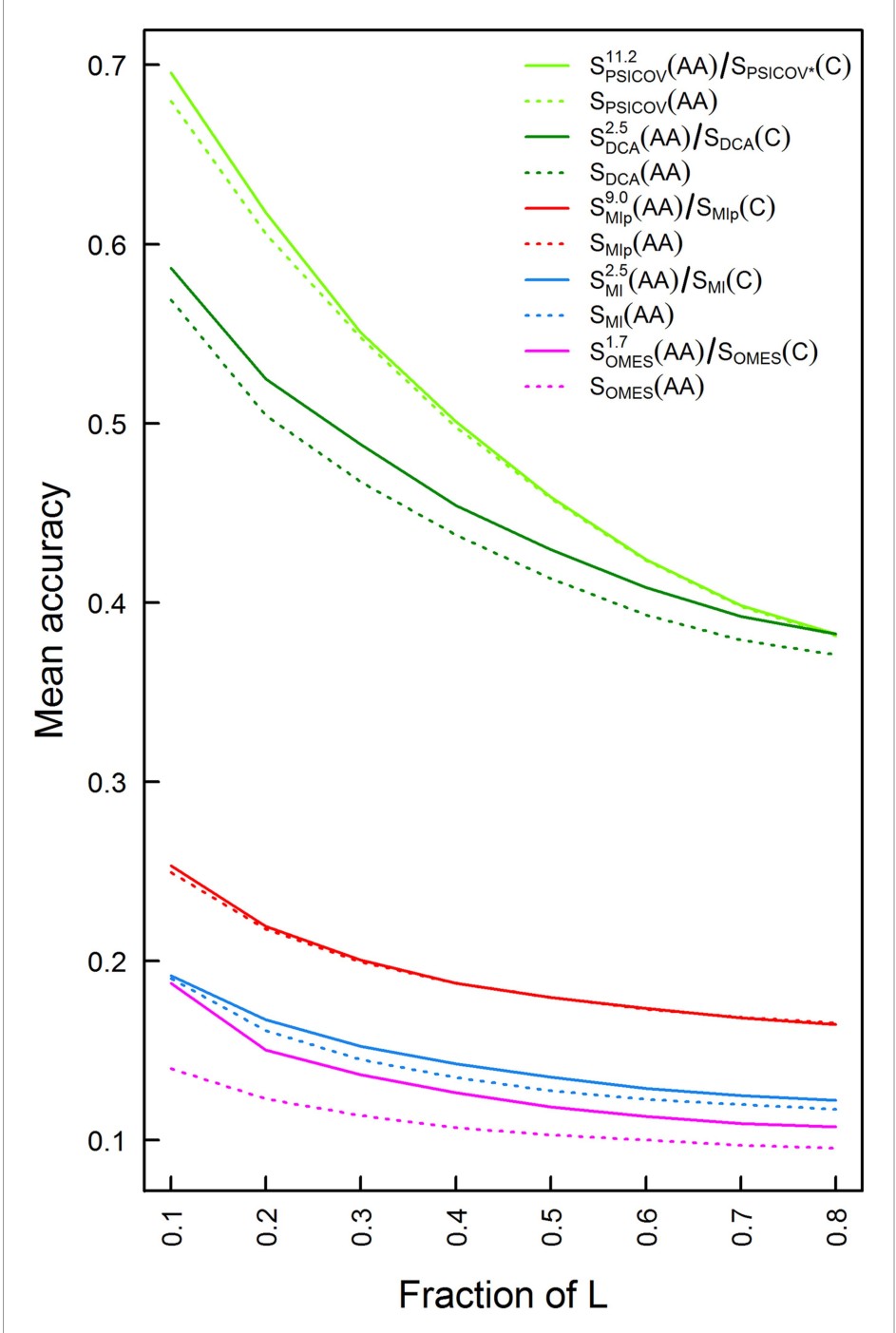

**Figure 2**. Plots of the mean accuracy of contact identification by various methods of correlated mutation analysis as a function of the top ranked fraction of protein length, L, number of predicted pairwise contacts. The mean accuracies of contact identification by the OMES, MI, MIp, DCA and PSICOV methods are shown either with or without incorporating codon data. Residues were defined as being in contact if the distance between their $C_\beta$ atoms is $\leq 8$ Å. PSICOV* indicates that it was carried out without the APC.

The following figure supplements are available for figure 2:

**Figure supplement 1**. Plots of the mean accuracy of contact identification by various methods of correlated mutation analysis as a function of the top ranked number of predicted pairwise contacts.

*Figure 2. continued on next page*

*Figure 2. Continued*

**Figure supplement 2**. Histogram of the fractions of residue pairs in physical contact out of those considered to be in contact according to two widely used definitions.

before (*Mao et al., 2015*). They also show that combining amino acid and codon data leads to an improvement in the predictions by OMES, MI, DCA and PSICOV. In the case of MIp, however, no improvement is observed despite the fact that this method performs worse than DCA and PSICOV. In MIp, a term called average product correction (APC) is subtracted from the MI score for each pair of positions in order to reduce false positives. Removing this correction from PSICOV where it also exists and including the codon data yielded the best method (*Figure 2*). Hence, we can conclude that there is an overlap between the background noise reduced upon including the APC term and codon data and that including the latter can be more advantageous as we observe for PSICOV.

The extent of improvement increases with increasing values of the power $\alpha$ until a maximum is reached (*Figure 3A*) at a value of $\alpha$ that depends on the method used and different values of $\alpha_{max}$ were, therefore, chosen accordingly. Cross-validation by dividing the MSA data into training and test sets showed that the values of $\alpha_{max}$ are stable, that is, they do not vary depending on the set of MSAs (*Figure 3—figure supplement 1*). Given these values of $\alpha_{max}$, the significance of the extent of improvement was assessed by comparing for each MSA the accuracy of the contact predictions using the different methods with and without incorporating codon data. Significance levels were determined using two non-parametric tests: (i) the Wilcoxon signed-rank test, which takes into account both the number of MSAs for which the accuracy of the contact predictions increases upon incorporating codon data (e.g., 81 in the case of DCA) and the magnitude of the improvement; and (ii) the sign test, which only considers the number of MSAs with improved accuracy. The extent of improvement achieved by incorporating codon data was found to be highly significant as indicated by the p-values obtained using both tests (*Figure 3B*).

The improvement in the predictions upon combining amino acid and codon data, when residues are defined as being in contact if the distance between their $C_\beta$ atoms is $\leq 8$ Å, led us to examine whether direct contacts are identified better using this contact definition compared with an 'All' definition used by others (*Morcos et al., 2011*) according to which a contact exists if at least one inter-atomic distance between the residues is $\leq 8$ Å. A non-redundant set of 2481 proteins with an available crystal structure at a resolution better than 1.6 Å was compiled and all the residue pairs in each structure that are in contact according to these two definitions were identified. We then determined for each protein what is the fraction of the residue pairs in contact according to these two definitions that are actually in direct physical contact, that is, with a distance $<3.5$ Å between two of their respective heavy atoms. It should be noted that atoms can interact with each other even if the distance between them is larger than 3.5 Å via, for example, weak electrostatic interactions but pairs of atoms which are closer than 3.5 Å can always be considered as interacting. It may be seen that, on average, pairs in direct contact constitute only about 10% of the pairs in contact according to the 'All' definition and 30% of the pairs in contact according to the $C_\beta$-based definition (*Figure 2—figure supplement 2*). The better success of DCA in identifying contacts according to the $C_\beta$-based definition when amino acid and codon data are combined is, therefore, an important result since more pairs that are in true physical contact are identified in this way.

Our finding that the $C_\beta$-based definition of contacts is better than the 'All' definition but still poor (only 30% of the pairs defined as being in contact are in physical contact) prompted us to test the performance of our method for additional contact definitions. The mean of the extent of improvement in contact prediction for 114 domains (or 86 in the case of PSICOV) was, therefore, determined as a function of the distance that must exist between at least two $C_\beta$ atoms in different residues in order for them to be defined as being in contact. It may be seen that, in the cases of PSICOV, OMES and DCA, the maximum improvements in contact prediction upon combining amino acid and codon data are when these distances are about 5.5, 7 and 5.5 Å, respectively, and that, in the cases of DCA and OMES, the improvement decreases dramatically when this distance is $>\sim 10$ Å (*Figure 4*). In the case of MI, the extent of improvement upon combining amino acid and codon data is found to be relatively insensitive to the distance used to define a contact and is maximal when it is $\sim 4.5$ Å (*Figure 4*). These

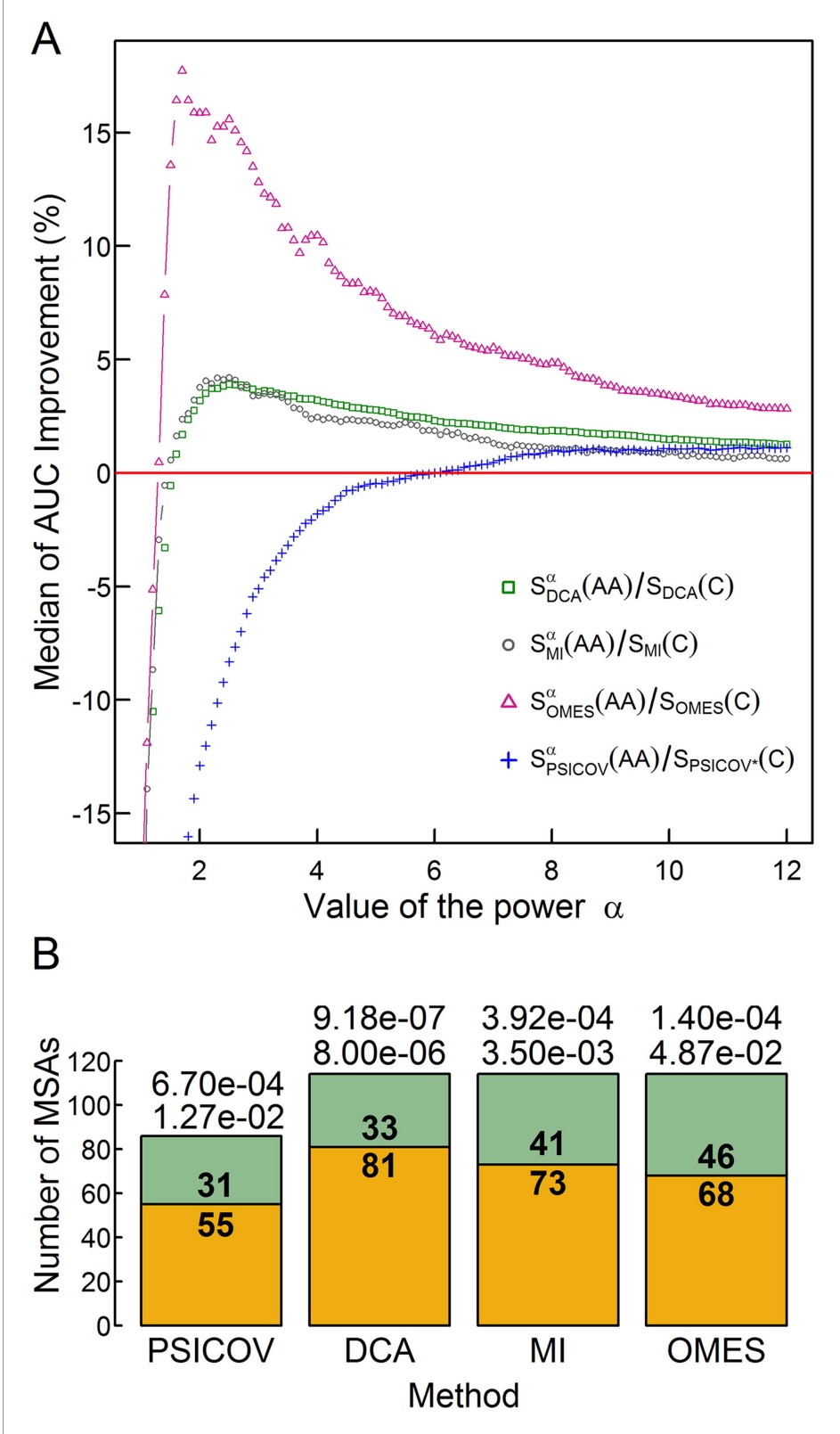

**Figure 3**. The effect of the relative weights of amino acid and codon information on contact prediction improvement and its statistical significance. (**A**) The median of the extent of improvement in contact prediction for 114 MSAs (86 in the case of PSICOV) is plotted as a function of the value of the power α which determines the relative weights of the

*Figure 3. Continued*

amino acid and codon correlations in the score, $S_i$ ($S_i = S_i^{\alpha}(AA)/S_i(C)$, where $S_i(AA)$ and $S_i(C)$ are the respective amino acid and codon scores generated by method i). The extent of improvement was determined by calculating the difference in the areas under the curves (AUC) of prediction accuracy vs number of predictions for each method i with and without incorporation of the codon data normalized by the area under the curve generated without codon data. The analysis was done for domains of length between 200 and 500 residues and at least 2000 coding sequences in their MSA. The value of $\alpha$ which maximizes the median improvement was used for predictions. Maximal respective improvements of 3.9% and 4.2% were found for DCA and MI when $\alpha$ is 2.5, 17.6% for OMES when $\alpha$ is 1.7 and 1.13% for PSICOV when $\alpha$ is 11.2. (**B**) Stacked bar plots showing the number of MSAs for which including codon data improved the contact predictions using the different methods (orange) and the number of those for which it was otherwise (green). The statistical significance of the improvement achieved by incorporating codon data is indicated by the top and bottom p-values obtained using the Wilcoxon signed-rank and sign tests, respectively.

The following figure supplement is available for figure 3:

**Figure supplement 1**. Testing the stability of the value of $\alpha$ by cross-validation.

data, therefore, show again that the improvement in contact prediction upon combining amino acid and codon data is greatest when the distance used for contact definition does not lead to many pairs being defined in contact when in fact they are not in direct physical contact.

The added value in combining amino acid and codon data can be illustrated for contact prediction by DCA in the case of Kex1Δp, a prohormone-processing carboxypeptidase from *Saccharomyces cerevisiae* that lacks the acidic domain and membrane-spanning portion of Kex1p. The crystal structure of Kex1Δp was solved at a resolution of 2.4 Å (*Shilton et al., 1997*) and its MSA consists of 1877 sequences. The predictions by DCA with or without incorporating codon data are shown in the respective top and bottom halves of the Kex1Δp contact map (*Figure 5A*). A comparison of the predictions by the two approaches shows that those made with incorporation of codon data are more long-range (in sequence) and more spread throughout the protein structure than those made without incorporation of codon data. Examples for such long-range contacts between different secondary structure elements in Kex1Δp that are predicted only when also the codon data is used include the interactions between Thr148 with Phe185, Ala186 with Leu208 and Leu190 with Leu368 (*Figure 5B*). This and other examples (*Figure 5—figure supplement 1*) show that incorporation of codon data can yield predictions of contacts between residues that are distant in sequence and are, thus, of more value for structure prediction.

## Conclusions

The input for methods for analysing correlated mutations has exclusively been multiple amino acid sequence alignments. Here, we have shown that improved contact prediction can be achieved by analysing both amino acid and codon MSAs together. The premise of our approach is that direct contacts are more likely if the correlation at the amino acid level is high but at the codon level is low. The score we propose, which reflects this expectation, can be used in conjunction with different methods of CMA but other possible scores should be examined in future work. Importantly, we find cases where contacts between residues that are distant in sequence and, thus, of greatest value for structure prediction are predicted only by using the combined method. Future work should test other potential applications of combined analysis of amino acid and codon MSAs such as predicting protein–protein interactions and, more generally, in feature selection in machine learning.

## Materials and methods

### Collection of sequences

Protein sequence datasets were collected from Pfam version 27.0 (*Finn et al., 2014*) based on representative proteomes (*Chen et al., 2011*) at 75% co-membership threshold (RP75). Protein coding sequences (CDS) of the collected proteins from Pfam were retrieved based on Uniprot cross reference annotations (for Refseq, Ensembl, EMBL and Ensembelgenomes databases in that order of priority) using the EMBL-EBI's WSDbfetch services (*McWilliam et al., 2009*) and Ensembl REST API

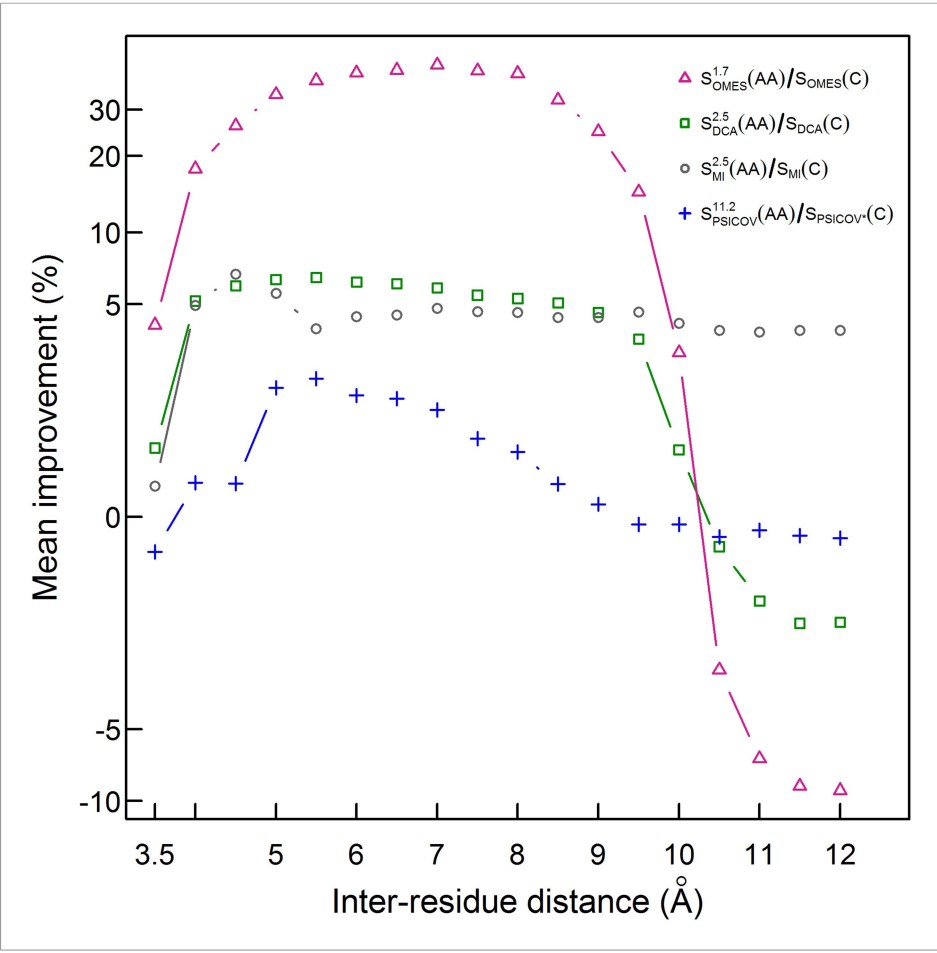

**Figure 4**. Improvement in contact prediction as a function of the distance used to define a physical contact. The mean of the extent of improvement in contact prediction for 114 domains (or 86 in the case of PSICOV) is plotted as a function of the distance that must exist between two $C_\beta$ atoms in different residues in order for them to be defined as being in contact. The extent of improvement was determined by calculating the difference in the areas under the curves of prediction accuracy vs number of predictions by OMES, MI, DCA and PSICOV with and without incorporation of the codon data normalized by the area under the curve generated without codon data. The analysis was done for domains of length between 200 and 500 residues and at least 2000 coding sequences in their MSA. The contact predictions were made for the seven sequences with available crystal structures that have the highest resolution and that in all cases is <3 Å.

(Beta version) (*Yates et al., 2015*). All collected CDSs were aligned in accordance to the Pfam HMM based MSAs using tranalign tool from the EMBOSS package (*Rice et al., 2000*). Pfam domain families with more than 2000 successfully retrieved coding sequences were used for further analysis (total of 551 MSAs). Only families with a known crystal structure at a resolution of 3 Å or better (more than 95% of the families have at least three such structures) and with an overlap of at least 80% of the domain sequence to the ATOM sequence in the solved structure were included in the analysis (total of 460 MSAs). Our analysis was also restricted for proteins with more than 200 residues that have a large number of potential contacts for prediction (114 MSAs). PDB structures were assigned to Pfam families in accordance to the mapping in the files downloaded from http://www.rcsb.org/pdb/rest/hmmer?file=hmmer_pdb_all.txt and ftp://ftp.ebi.ac.uk/pub/databases/msd/sifts/text/pdb_chain_uniprot.lst. PDB structures were retrieved and their coordinates were extracted using the bio3D R package (*Grant et al., 2006*). Pairwise sequence alignments for mapping were performed using Biostrings (Pages H., Aboyoun P., Gentleman R. and DebRoy S. Biostrings: String objects representing biological sequences, and matching algorithms. R package version 2.34.1).

## Evaluation of prediction accuracy

The evaluation was based on the all structures with the highest resolution (at least 3 Å) but, in cases where families have more than 30 known structures with unique sequences, only 30 with the best resolution were used (in cases of structures with the same resolution we arbitrarily chose one). The average accuracy of contact predictions for all the crystal structures of each domain family was then calculated so that domain families with many crystal structures would not be over-represented. Two definitions for a contact between two amino acids were employed: a distance of less than 8 Å between $C_\beta$ atoms and a distance of less than 8 Å between any two heavy atoms. Only pairs of residues that are separated by at least five amino acids in the protein sequence were considered. Accuracy was calculated as the proportion of true contacts from the N pairs with the highest score in that set. We evaluated the improvement of our method using the difference in the area under the curve (AUC) of the accuracy vs number of predicted pairs of our method relative to the results of the original OMES, MI, MIp, PSICOV and DCA methods. AUC was calculated using the auc function in MESS package in R with the default parameters.

## Determination of the number of pairs in physical contact using different contact definitions

A non-redundant set of 2481 PDB entries with a percentage identity cutoff of 20%, resolution better than 1.6 Å and an R-factor cutoff of 0.25 was downloaded from the pre-compiled CullPDB lists (*Wang and Dunbrack, 2003*) at http://dunbrack.fccc.edu/Guoli/pisces_download.php on February 25, 2015. Two residues were defined to be in a physical contact if they have at least one pair of atoms with a distance ≤3.5 Å. The number of true physical contacts, that is, those with a distance ≤3.5 Å between two of their respective heavy atoms, was determined for each protein in the set and divided by the number of residue pairs defined to be in contact if at least one inter-atomic distance between them is ≤8 Å (designated 'All') or if the distance between their $C_\beta$ atoms is ≤8 Å. Only pairs of residues that are separated by at least five amino acids along the protein sequence were considered.

## Methods for analysing correlated mutations

The score for a pair of positions i and j, S (i,j), for the OMES (Observed Minus Expected Squared)

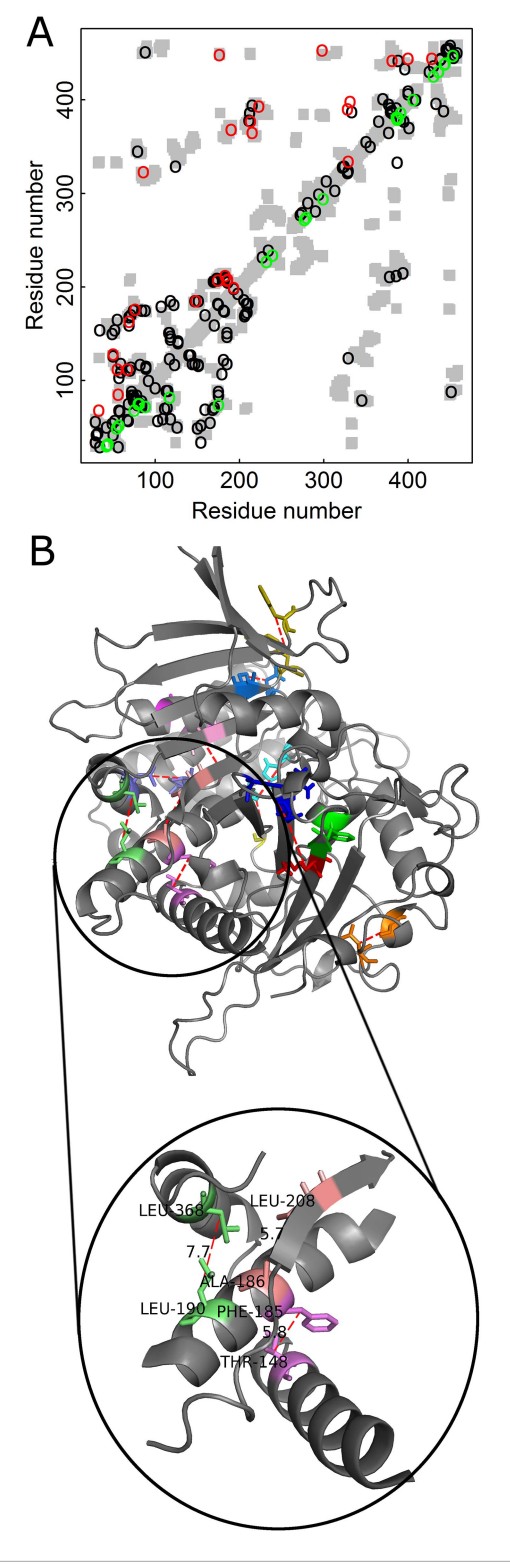

**Figure 5**. Added value of combining amino acid and codon data in contact prediction by DCA illustrated for Kex1Δp, a prohormone-processing carboxypeptidase from *Saccharomyces cerevisiae*. (**A**) Contact map of the structure of Kex1Δp (PDB ID: 1AC5) in which all the
*Figure 5. continued on next page*

*Figure 5. Continued*

contacts are shown as gray rectangles. Residues were defined as being in contact if the distance between their $C_\beta$ atoms ($C_\alpha$ for glycine) is ≤8 Å. The top 100 predictions of contacts made with or without incorporating codon data are highlighted above (in red) and below (in green) the diagonal, respectively, and those predicted by both methods by black circles. (**B**) The crystal structure of Kex1Δp with predicted contacts highlighted. Only true predicted contacts that were not predicted by the original method are highlighted. Each contacting pair has a different color. The contacts were predicted using an MSA with 1877 coding sequences with a length of 415 codons. The magnified region shows some long-range contacts between different secondary structure elements that are predicted only when also the codon data is used.

The following figure supplement is available for figure 5:

**Figure supplement 1**. Illustration for four proteins of added value of combining amino acid and codon data in contact prediction by DCA.

method was calculated, as follows (*Kass and Horovitz, 2002*; *Fodor and Aldrich, 2004*):

$$S_{OMES}(i,j) = \sum_a \sum_b \left( OBS_{a_i b_j} - EXP_{a_i b_j} \right)^2$$

where $OBS_{a_i b_j}$ and $EXP_{a_i b_j}$ are the respective observed and expected number of sequences in the MSA with residue type a at position i and residue type b at position j. The score for the mutual information, MI, method was calculated as follows (*Gloor et al., 2005*):

$$S_{MI}(i,j) = \sum_{a=1}^{21} \sum_{b=1}^{21} f_{(i,a:j,b)} \, log \frac{f_{(i,a:j,b)}}{f_{(i,a)} f_{(j,b)}}$$

where $f_{(i,a)}$ and $f_{(j,b)}$ denote the respective frequencies of occurrence of residue type a at position i and residue type b at position j and $f_{(i,a;\,j,b)}$ denotes the joint probability of occurrence of residue type *a* at position *i* and type *b* at position *j*. In the case of the MIp method (*Dunn et al., 2008*), an average product correction (APC) term is subtracted from the MI score for each pair of positions. The APC term, which is a measure of the background MI shared by positions i and j, is given by:

$$APC(i,j) = \frac{MI_{(i,\overline{x})} MI_{(j,\overline{x})}}{\overline{MI}}$$

where terms in the nominator are the respective average MI values of positions i and j with all other positions in the alignment and the term in the denominator is the average background MI of all the positions in the alignment. The MIp score is given by:

$$S_{MI_p}(i,j) = S_{MI}(i,j) - APC(i,j)$$

The Direct Coupling Analysis (DCA) method (*Morcos et al., 2011*) was implemented in R for amino acid and codon MSAs based on a Matlab source code provided by Weigt et al. (http://dca.rice.edu/portal/dca/download). The PSICOV code was downloaded from http://bioinfadmin.cs.ucl.ac.uk/downloads/PSICOV/ and used for the predictions based on amino acid MSAs with the default parameters for faster options as recommended by the authors (-p -r 0.001 and with the -l option in order to avoid using the APC term). The PSICOV code was modified in order to carry out the same analysis for codon MSAs and a python script was implemented to perform the whole analysis as done for the other methods using Pfam MSA files in Stockholm format and fasta MSA files as inputs. PSICOV was used here either with the APC for amino acid MSAs or without the APC for the predictions based on both amino acid and codon MSAs.

## Available software

The R and *Python* source codes for the contact prediction by all methods, C source code modifications to PSICOV V2.1b3, R source code for structure-domain sequence mapping and python scripts for generating codon MSAs are available at https://etaijacob.github.io/. Details on the relevant R packages that will be available on CRAN will also be provided at: https://etaijacob.github.io/.

## Acknowledgements

This work was supported by grants 158/12 (to A. H.) and 772/13 (to R. U.) of the Israel Science Foundation. A.H. is an incumbent of the Carl and Dorothy Bennett Professorial Chair in Biochemistry.

## Additional information

### Funding

| Funder | Grant reference | Author |
|---|---|---|
| Israel Science Foundation (ISF) | 772/13 | Ron Unger |
| Israel Science Foundation (ISF) | 158/12 | Amnon Horovitz |

The funders had no role in study design, data collection and interpretation, or the decision to submit the work for publication.

### Author contributions

EJ, Conception and design, Acquisition of data, Analysis and interpretation of data, Drafting or revising the article; RU, AH, Conception and design, Analysis and interpretation of data, Drafting or revising the article

## Additional files

### Major datasets

The following previously published datasets were used:

| Author(s) | Year | Dataset title | Dataset ID and/or URL | Database, license, and accessibility information |
|---|---|---|---|---|
| Finn RD, Bateman A, Clements J, Coggill P, Eberhardt RY, Eddy SR, Heger A, Hetherington K, Holm L, Mistry J, Sonnhammer EL, Tate J, Punta M | 2013 | Pfam version 27.0 | http://pfam.xfam.org/ | All domain families at RP75 redundancy level are publicly available at Pfam EMBL-EBI ftp site. |
| Pruitt KD, Brown GR, Hiatt SM, Thibaud-Nissen F, Astashyn A, Ermolaeva O, Farrell CM, Hart J, Landrum MJ, McGarvey KM, Murphy MR, O'Leary NA, Pujar S, Rajput B, Rangwala SH, Riddick LD, Shkeda A, Sun H, Tamez P, Tully RE, Wallin C, Webb D, Weber J, Wu W, Dicuccio M, Kitts P, Maglott DR, Murphy TD, Ostell JM | 2015 | RefSeq | http://www.ncbi.nlm.nih.gov/refseq/ | All RefSeq sequences are publicly available at NCBI Reference sequence database using ftp service or EMBL-EBI's WSDbfetch services. |
| Cunningham F, Ridwan Amode M, Barrell D, Beal K, Billis K, Brent S, Carvalho-Silva D, Clapham P, Coates G, Fitzgerald S, Gil L, Girón CG, Gordon L, Hourlier T, Hunt SE, Janacek SH, Johnson N, Juettemann T, Kähäri AK, Keenan S, Martin FJ, Maurel T, McLaren W, Murphy DN, Nag R, Overduin B, Parker A, Patricio M, Perry E, Pignatelli M, Riat HS, Sheppard D, Taylor K, Thormann A, Vullo A, Wilder SP, Zadissa A, Aken BL, Birney E, Harrow J, Kinsella R, Muffato M, Ruffier M, Searle SMJ, Spudich G, Trevanion SJ, | 2015 | Ensembl | http://www.ensembl.org/ | All Ensemblgenomes sequences are publicly available at ENSEMBLGENOMES site using ftp or REST API. |

| Author(s) | Year | Dataset title | Dataset ID and/or URL | Database, license, and accessibility information |
|---|---|---|---|---|
| Yates A, Zerbino DR, Flicek P | | | | |
| Berman HM, Westbrook J, Feng Z, Gilliland G, Bhat TN, Weissig H, Shindyalov IN, Bourne PE | 2015 | Protein Data Bank | http://www.rcsb.org/ | All pdbcodes in this work are publicly available at RCSB Protein Data Bank using ftp. |
| Wang G, Dunbrack RL | 2015 | CullPDB | http://dunbrack.fccc.edu/Guoli/culledpdb/ | All datasets are publicly available at PISCES server home page. |
| Velankar S, Dana JM, Jacobsen J, van Ginkel G, Gane PJ, Luo J, Oldfield TJ, O'Donovan C, Martin MJ, Kleywegt GJ | 2015 | SIFTS | ftp://ftp.ebi.ac.uk/pub/databases/msd/sifts/text/ | All datasets are publicly available at ENSEMBL site. |
| The UniProt Consortium | 2014 | Uniprot | http://www.uniprot.org/ | All uniprot entriess are publicly available at uniprot site. |
| Leinonen R, Akhtar R, Birney E, Bower L, Cerdeno-Tárraga A, Cheng Y, Cleland I, Faruque N, Goodgame N, Gibson R, Hoad G, Jang M, Pakseresht N, Plaister S, Radhakrishnan R, Reddy K, Sobhany S, Ten Hoopen P, Vaughan R, Zalunin V, Cochrane G | 2015 | European Nucleotide Archive | http://www.ebi.ac.uk/ena | All sequences are publicly available at EBI ENA site using ftp service or EMBL-EBI's WSDbfetch services. |
| Kersey PJ, Allen JE, Christensen M, Davis P, Falin LJ, Grabmueller C, Hughes DS, Humphrey J, Kerhornou A, Khobova J, Langridge N, McDowall MD, Maheswari U, Maslen G, Nuhn M, Ong CK, Paulini M, Pedro H, Toneva I, Tuli MA, Walts B, Williams G, Wilson D, Youens-Clark K, Monaco MK, Stein J, Wei X, Ware D, Bolser DM, Howe KL, Kulesha E, Lawson D, Staines DM | 2015 | Ensemblegeneomes | http://ensemblgenomes.org/ | All Ensemblgenomes sequences are publicly available at ENSEMBLGENOMES site using ftp or REST API. |

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
