## [Decision Letter]

Thank you for submitting your work entitled “Codon-level information improves predictions of inter-residue contacts in proteins by correlated mutation analysis” for peer review at *eLife*. Your submission has been favorably evaluated by Aviv Regev (Senior Editor) and two reviewers, one of whom is a member of our Board of Reviewing editors. The Reviewing Editor has drafted this decision to help you prepare a revised submission.

The paper presents a potentially important advance in the area of deducing spatial contacts between amino acids in globular proteins, by proposing a simple yet elegant modification, which combines the analysis of amino acid and codon MSAs. While the results are potentially interesting, there are some important statistical analyses which are still lacking and need to addressed in the revision. In particular (1) the authors should incorporate statistical testing to assess the significance of the differences in accuracy with and without incorporating codon information. One would want to know if the improvement is robust, and it could be also that there are specific sub-classes where it is more/less beneficial. The scope of alternative methods to compare to should also be expanded. (2) The authors should show compelling analyses that address the question of over-fitting vs. generalization. This can be accomplished by cross-validation (the correct way to choose a set of parameters), and then with an additional test set (for seeing how the parameters chosen by cross validation perform on new data).

Essential revisions:

1) In Figure 1, there are differences shown for accuracy with and without incorporating codon information, but it is unclear if these differences are significant. Figure 1 shows the mean accuracy across the alignments, but it would be straightforward to also show or report the variance or standard deviation across the alignments. Then an explicit P-value could be put on a null hypothesis that incorporation of codon information increases or decreases accuracy. In general, the apparent result in Figure 1 that incorporation of codon information into S_DCA_ either helps or hurts depending on how contacts are defined reduces confidence that incorporation of codon information is robustly improving the underlying algorithms.

2) In Figure 1, the incorporation of codon information appears to help OMES and MI, but a more appropriate comparison might be to other algorithms that attempt to correct for phylogenetic artifacts (see for example http://bioinformatics.oxfordjournals.org/content/24/3/333.short; Mutual information without the influence of phylogeny or entropy dramatically improves residue contact prediction).

3) The data in Figure 2 should be on one plot. The figure needs to clearly emphasize that one set of lines are without the codon data and the other set are with it. This makes a huge difference and it the important conclusion of the work. The effect of how close distances are determined (C_β_ or All) is minor and confusing.

4) In tuning a free parameter, such as is being performed in Figure 3, it is not surprising that a value for the free parameter alpha can be found that improves the performance. But does this just reflect over-fitting? It would perhaps be more informative to split the data into a training set and a test set, and evaluate the improvement on the test set which was not used to estimate alpha. By repeating this process on permutations of randomly chosen test sets and training sets, it might be possible to determine how stable estimates of alpha are or, alternatively, to demonstrate that the optimal value of alpha is highly variable from training set to training set.

---

## [Author Response]

*Essential revisions*:

*1) In*
Figure 1*, there are differences shown for accuracy with and without incorporating codon information, but it is unclear if these differences are significant.*
Figure 1
*shows the mean accuracy across the alignments, but it would be straightforward to also show or report the variance or standard deviation across the alignments. Then an explicit P-value could be put on a null hypothesis that incorporation of codon information increases or decreases accuracy. In general, the apparent result in*
Figure 1
*that incorporation of codon information into S*_*DCA*_
*either helps or hurts depending on how contacts are defined reduces confidence that incorporation of codon information is robustly improving the underlying algorithms*.

We agree with the reviewers that an assessment of the significance of our results should have been provided. We have, therefore, added a new panel to Figure 3 that shows stacked bar plots of the number of MSAs for which including codon data improved the contact predictions using the different methods and the number of MSAs for which it was otherwise. The improvement achieved by incorporating codon data is shown to be significant as indicated by P-values obtained using the Wilcoxon signed-rank and sign tests.

*2) In*
Figure 1*, the incorporation of codon information appears to help OMES and MI, but a more appropriate comparison might be to other algorithms that attempt to correct for phylogenetic artifacts (see for example http://bioinformatics.oxfordjournals.org/content/24/3/333.short; Mutual information without the influence of phylogeny or entropy dramatically improves residue contact prediction)*.

In the original manuscript, we showed that incorporation of codon information improves predictions by the OMES, MI and DCA methods. Following the comment of the reviewers, we tested whether incorporation of codon information improves predictions also by the MIp method and were surprised to see no significant improvement. In MIp, the background MI shared by positions i and j (termed APC) is subtracted from their MI score. It appears, therefore, that there is an overlap between the improvements generated by including the APC term and codon data. The PSICOV method, which appears to outperform all other methods, also includes the APC term. We, therefore, tested whether including codon data instead of the APC term improves the PSICOV method and found that there is a significant improvement. The effects of including codon data on the predictions by all 5 methods (OMES, MI, MIp, DCA and PSICOV) are shown in the revised Figure 2. Although incorporation of codon data does not improve the MIp method, it does lead to an improvement in the two state-of-the-art methods, DCA and PSICOV, which outperform all the older methods including MIp.

*3) The data in*
Figure 2
*should be on one plot. The figure needs to clearly emphasize that one set of lines are without the codon data and the other set are with it. This makes a huge difference and it the important conclusion of the work. The effect of how close distances are determined (C*_*β*_
*or All) is minor and confusing*.

Figure 2 was altered as suggested by the reviewers so that the comparison of all the methods with and without incorporating codon data is shown in the same figure using only the C_β_ definition which is better at identifying contacts and is the most widely used. The comparison between C_β_ and All has been removed as we agree that it is minor and confusing.

*4) In tuning a free parameter, such as is being performed in*
Figure 3*, it is not surprising that a value for the free parameter alpha can be found that improves the performance. But does this just reflect over-fitting? It would perhaps be more informative to split the data into a training set and a test set, and evaluate the improvement on the test set which was not used to estimate alpha. By repeating this process on permutations of randomly chosen test sets and training sets, it might be possible to determine how stable estimates of alpha are or, alternatively, to demonstrate that the optimal value of alpha is highly variable from training set to training set*.

We followed the reviewers’ suggestion and split the MSAs into 1000 different training and test tests. The alpha was determined for each learning set and then used for predictions carried out for its corresponding test set. The results, which are presented in a new supplementary Figure (Figure 3—figure supplement 1), show that the estimates of alpha are indeed stable and that the prediction accuracies for the learning and test sets are similar.